# Impact of *GSTA1* Polymorphisms on Busulfan Oral Clearance in Adult Patients Undergoing Hematopoietic Stem Cell Transplantation

**DOI:** 10.3390/pharmaceutics11090440

**Published:** 2019-09-01

**Authors:** Veronique Michaud, My Tran, Benoit Pronovost, Philippe Bouchard, Sarah Bilodeau, Karine Alain, Barbara Vadnais, Martin Franco, François Bélanger, Jacques Turgeon

**Affiliations:** 1Faculty of Pharmacy, Université de Montréal, Montreal, QC H3C 3J7, Canada; 2CRCHUM, Centre de Recherche du Centre Hospitalier de l’Université de Montréal, Montreal, QC H2X 0A9, Canada; 3College of Pharmacy, Lake Nona Campus, University of Florida, Orlando, FL 32827, USA; 4Hôpital Maisonneuve-Rosemont, Montreal, QC H1T 2M4, Canada

**Keywords:** busulfan, glutathione S-transferase, genetic polymorphism, limited sampling strategy, pharmacokinetics

## Abstract

Background: Busulfan pharmacokinetics exhibit large inter-subject variability. Our objective was to evaluate the influence of glutathione S-transferase A1 (*GSTA1*) gene variants on busulfan oral clearance (CLo) in a population of patients undergoing hematopoietic stem cell transplantation. Methods: This is a quasi-experimental retrospective study in adult patients (*n* = 87 included in the final analyses) receiving oral busulfan. Pharmacokinetics data (area under the plasma concentration-time curve (AUC) determined from 10 blood samples) were retrieved from patients’ files and *GSTA1 *A* and **B* allele polymorphisms determined from banked DNA samples. Three different limited sampling methods (LSM) using four blood samples were also compared. Results: Carriers of *GSTA1*B* exhibited lower busulfan CLo than patients with an **A/*A* genotype (*p* < 0.002): Busulfan CLo was 166 ± 31, 187 ± 37 vs. 207 ± 47 mL/min for *GSTA1*B/*B,*
**A/*B* and **A/*A* genotypes, respectively. Similar results were obtained with the tested LSMs. Using the standard AUC method, distribution of patients above the therapeutic range after the first dose was 29% for *GSTA1*A/*A*, 50% for **A/*B,* and 65% for **B/*B*. The LSMs correctly identified ≥91% of patients with an AUC above the therapeutic range. The misclassified patients had a mean difference less than 5% in their AUCs. Conclusion: Patients carrying *GSTA1* loss of function **B* allele were at increased risk of overdosing on their initial busulfan oral dose. Genetic polymorphisms associated with *GSTA1* explain a significant part of busulfan CLo variability which could be captured by LSM strategies.

## 1. Introduction

In current hematopoietic stem cell transplantation (HSCT) practices, busulfan is a commonly used alkylating agent. When combined with other drugs, busulfan exhibits a beneficial immunosuppressive effect [1]. The drug has a very narrow therapeutic index which requires close therapeutic monitoring. Low concentrations of busulfan can result in an increased risk of graft failure and recurrence of the disease whereas high concentrations of busulfan can result in an increased risk of hepatic toxicity [2,3]. Current therapeutic monitoring methods of the drug involve taking numerous (often up to 10) blood samples to calculate patient’s plasma concentration vs. time area under the curve (AUC) [4]. However, we and others have demonstrated the value of limited sampling strategies to estimate mean busulfan plasma concentration and compute required busulfan doses in these leukemic patients [5,6,7,8,9].

The glutathione S-transferase enzymes (GSTs) are important Phase II biotransformation enzymes that catalyze the conjugation of many hydrophobic and electrophilic compounds with reduced glutathione [10,11]. Based on their biochemical, immunologic, and structural properties, soluble GSTs (including cytosolic and mitochondrial forms) are divided into several classes; alpha, mu, kappa (mitochondrial), theta, pi, omega, and zeta [10,11]. The GST alpha 1 (A1) isoform is mainly expressed in the liver, intestine, kidneys and endocrine tissues and contributes to the metabolism of several anticancer drugs as well as steroids and products of lipid degradation [12,13]. The *GSTA1* gene has been mapped to the GST-alpha gene cluster on chromosome 6p12, it is approximately 12 kb long and contains seven exons [14]. *GSTA1* expression is influenced by a genetic polymorphism that consists of two alleles, *GSTA1*A* and *GSTA1*B*, containing three linked base substitutions in the proximal promoter, at positions −567, −69, and −52 [14,15]. The *G*-to-*A* change at position −52 appears to be responsible for the differential promoter activities of *GSTA1*A* and *GSTA1*B*, expression of *GSTA1*A* being greater than *GSTA1*B*. 

Busulfan pharmacokinetics properties are highly variable among patients and dosing regimens are affected by patients’ characteristics such as body weight, age and genotype [16]. For instance, busulfan pharmacokinetics in children differs largely from that observed in adults as clearance decreases with age even when expressed relative to body weight or body surface area [17]. Notably, busulfan is a lipophilic molecule with highly variable absorption and bioavailability [18]. The drug is highly protein bound and extensively metabolized in the liver with less than 2% being excreted unchanged in the urine [19,20]. Busulfan is mainly metabolized through conjugation with glutathione by the major hepatic isoform *GSTA1*. In vitro experiments showed that two other isoenzymes, *GSTM1* and *GSTP1*, contribute to a lesser extent in the formation of busulfan glutathione conjugates (46% and 18% of *GSTA1* busulfan activity, respectively) [19]. At this time, the relevance of *GSTA1* polymorphisms on busulfan pharmacokinetics in adults, following oral administration, has been suggested but not clearly established [16,21,22,23,24,25].

The primary objective of our study was to investigate the influence of *GSTA1* gene variants on busulfan oral clearance in adult patients. Our secondary objective was to combine use of genetic information and AUCs calculated from various limited sampling models (LSM) to characterize the predictive value of these joint strategies for required oral busulfan dose.

## 2. Methods

This is a quasi-experimental retrospective study. De-identified pharmacokinetic data generated in the context of a standard of care procedure was collected from adult patients who underwent HSCT preparation at Maisonneuve-Rosemont hospital over a 4-year period. The research protocol was approved by the ethics committee of Maisonneuve-Rosemont hospital (No. 06068; 5 October 2006). 

### 2.1. Clinical Study Design

Adult patients (*n* = 119) aged 18 years and older receiving an oral dose of busulfan 4 mg/kg/d (using ideal body weight) divided into 4 doses per day for 4 days (total of 16 doses) were included in this study. Patients were excluded if they vomited in the hour following administration of the first dose. Patients who vomited and who required the administration of additional busulfan tablets were also excluded. Patients were also excluded if a complete pharmacokinetic profile could not be generated or if a DNA sample for genotype determination could not be obtained (e.g., patient’s refusal to participate in Maisonneuve-Rosemont DNA banking for research purposes). A total of 97 pharmacokinetic profiles were obtained following the first administration of busulfan or after the second dose for 3 patients (therapeutic monitoring could not be performed on the first dose and standard dose was administered on first and second dose). Standard therapeutic drug monitoring consisted of obtaining 10 blood samples drawn at 0, 20, 40, 60, 90, 120, 180, 240, 300 and 360 min following the first busulfan dose on day one. Additional therapeutic drug monitoring was performed on subsequent doses in patients for whom the dose of busulfan was modified based on their pharmacokinetic profile (target AUC at Maisonneuve-Rosemont hospital = 1150–1450 μmol·min/L; 283, 245–357, 140 ng·min/mL). 

### 2.2. Pharmacokinetic Profile Determination

Pharmacokinetic profiles were obtained by reviewing medical charts. Busulfan plasma levels were determined by a validated HPLC assay with UV detection [26]. The drug concentration–time data were analyzed by standard noncompartmental methods using WinNonLin^®^ 10.0 software (Certara, Mountain View, CA, USA) to determine AUC_0→∞_(considered as the reference AUC). Apparent oral clearance (CLo) of busulfan was calculated as CL/F = Dose_(oral)_/AUC_0→∞ (oral)_.

### 2.3. Genotyping Procedure

*GSTA A1 C<-69>T* polymorphism was determined by polymerase chain reaction-restriction fragment length polymorphism as described by Kusama et al. with minor modifications. [24] A 821 bp fragment in the promoter region of the *GSTA1* gene was amplified with a forward primer (F: *5′-CCC TAC ATG GTA TAG GTG AAA T-3′*) and reverse primer (R: *5′-GTG CTA AGG ACA CAT ATT AGC-3′*). PCR reactions were performed in a PTC-100 Thermal Cycler (MJ Research Inc., Watertown, MA, USA) under the following conditions: an initial 5 min denaturation step at 95 °C, followed by 35 cycles of 1 min for each step i.e., denaturation at 96 °C, annealing at 63 °C and extension at 72 °C, and a final extension step at 72 °C for 5 min. PCR products were digested with H*inf*I for 3–4 h at 37 °C and separated by electrophoresis (100 V, 45 min) on a 2% agarose/Synergel.

### 2.4. Validation Cohort

Genotyping procedures for *GSTA1* were also performed in random samples (*n* = 116) obtained from a genetic bank constituted of isolated DNA samples provided by a group of individuals (18–25 years old) without known cardiovascular diseases. These analyses were performed to establish *GSTA1* allele frequencies in “young heathy” adults. Consent was obtained from each individual prior to participation in this DNA banking initiative.

### 2.5. Comparison of the Standard Sampling Strategy to LSMs

We compared results of the standard sampling model to LSMs. From our previous paper, we have determined that the Bullock 4 limited sampling model as well as the New 4.2 and the New 4.3 LSM would be ideal for this study [5,6]. The Bullock 4 LSM requires blood samples at 0.5, 1, 4, and 6 h after the first dose whereas the New 4.2 LSM require blood samples at 1, 1.5, 3, and 6 h after the first dose while New 4.3 LSM requires blood samples at 1, 2, 4 and 6 h post-dose.

### 2.6. Statistical Analyses

Data are expressed as mean ± SD. The AUCs and oral clearance of busulfan were compared across the genotype groups of *GSTA1* using non-parametric tests. Tukey correction was used to determine the *p* values for multiple comparisons. The allele and genotype frequencies, and Hardy-Weinberg equilibrium were analyzed. Statistical analyses were performed using GraphPad v7.05 (GraphPad Software, Inc., San Diego, CA, USA).

## 3. Results

Over the four-year period of our study, 119 patients received oral busulfan. Therapeutic monitoring was performed on the first (or second dose, *n* = 3) of busulfan. A total of 100 pharmacokinetic profiles were obtained from those patients’ medical charts. Genetic analyses were performed in 89 patients of which two patients were excluded (DNA quality). The characteristics of the 87 patients included in our final analysis are presented in Table 1. Fifty-five percent (55%) of these patients were male. Mean age was 48.3 ± 9.7 (range 25–65) years, adjusted body weight was 65.2 ± 10 (range 46–88) kg, and their lean body weight was 63.3 ± 9.6 (44–84) kg. Acetaminophen, which could decrease glutathione reserve, was co-administered in 23 patients. Antifungals such as voriconazole and fluconazole but not itraconazole (which has been associated with a decrease in busulfan clearance) were co-administered in six patients (*n* = 1 and 5, respectively). The mean initial dose of busulfan administered was 65 mg and the mean population AUC was 358,066 ng·min/mL. 

The genotype frequencies found in our cohort were 27.5% (*n* = 24), 45.9% (*n* = 40), and 26.4% (*n* = 23) for the *GSTA1*A/*A*, **A/*B*, and **B/*B* groups, respectively. These frequencies were in Hardy-Weinberg equilibrium but differ from the distribution of alleles observed in our validation cohort (Table 2); more patients presented with a **B*B* genotype (26.4%) compared to young healthy subjects (20%). Demographic data among *GSTA1* genotype groups are presented in Table 1. There was no significant difference observed in most of these parameters among the groups except for alkaline phosphatase (APL) and lactate dehydrogenase (LDH) levels. The difference observed for the LDH results can be explained by outlier values for two individuals in the *GSTA1*A*A* group. A higher proportion of patients receiving acetaminophen was found in the *GSTA1*A*A* group. However, there was no statistically significant difference in measured AUC or in the apparent oral clearance of busulfan between acetaminophen users and non-users (*p* = 0.6).

Pharmacokinetic profiles obtained from patients demonstrated that 33/87 (38%) patients reached therapeutic range on the first dose: 12 patients were exhibiting subtherapeutic levels while 42 patients were having supratherapeutic levels. Figure 1 illustrates that higher AUCs were observed in patients with a *GSTA1*B*B* genotype (395,562 ± 77,083 ng/mL/min) compared to *GSTA1*A/*B* (357,062 ± 53,100 ng/mL/min) and *GSTA1*A/*A* patients (323,691 ± 65,906 ng/mL/min; *p* < 0.001). Hence, carriers of *GSTA1*B* (*n* = 64) were significantly associated with lower busulfan CLo compared to wild-type *GSTA1*A*: 179 ± 36 vs. 207 ± 47 mL/min (*p* = 0.003). Busulfan CLo among the three genotype groups are illustrated in Figure 2: 166 ± 31, 187 ± 40 and 207 ± 47 mL/min, for *GSTA1*B/*B*, **A/*B* and **A/*A*, respectively.

Using the standard AUC method, distribution of patients (%) above the therapeutic range after the first dose was 29% for *GSTA1*A/*A*, 50% for **A/*B* and 65% for **B/*B* (Figure 3). Patients with a *GSTA1*A/*A* genotype were more likely to have achieved therapeutic levels (overall 42%) after the first dose of treatment compared to subjects with a *GSTA1*B/*B* genotype (26%).

The LSMs correctly associated 91% of patients with their therapeutic level category. In our final patients’ cohort (*n* = 87), percent of patients with busulfan mean concentrations in the therapeutic range were 38%, 37%, 38% and 41% for the standard model (AUC with 10 time points), Bullock 4 model, New 4.2 and New 4.3 models, respectively (Appendix A). Patients with busulfan mean concentrations above the therapeutic range were 48%, 47%, 44% and 44% for the standard model, Bullock 4 model, New 4.2 and New 4.3 models, respectively. The misclassified patients had a mean difference less than 5% (±4.8%, range AUC_ref_/AUC_LSM_ 0.89–1.05) in their AUCs. The proportion of patients and their corresponding therapeutic levels using LSMs is illustrated in Figure 4 for the three *GSTA1* genotype groups. The LSMs correctly identified busulfan’s AUC above the therapeutic range for individuals carrying *GSTA1*B*B* genotype for 15/15 (100%) using the Bullock 4 model and for 14/15 using New 4.2 and New 4.3 models. The only misclassified patient had a difference of 6% in the estimated AUCs compared to the standard AUC determination model.

## 4. Discussion

In this study, we demonstrated that the administration of an initial standard oral dose of busulfan (1 mg/kg of a 4 mg/kg/day regimen) to patients with a *GSTA1*B*B* genotype was associated with higher plasma concentrations of busulfan and consequently, with lower estimated oral clearance of the drug. More patients with a *GSTA1*B*B* genotype were exhibiting mean plasma concentrations above the targeted therapeutic range for busulfan after the initial dose which could predispose them to increased toxicity from the drug. We also demonstrated that patients from the various *GSTA1* genotypes could be efficiently classified for their therapeutic level status by limited sampling strategies using four blood samples instead of 10. 

Busulfan pharmacokinetics has been the subject of intense research due to important inter-subject variability and its narrow therapeutic index [2,3,16]. Clinical consequences of inappropriate dosing are well established with significant loss of efficacy in patients with sub-therapeutic levels and toxicity in patients with supra-therapeutic levels of the drug [2]. Various determinants of busulfan pharmacokinetics have been identified including weight, age and genetics. Dosing based on lean body weight and dose adjustment with age are well established [17]. However, the role of genetic polymorphisms still remains to be confirmed.

In the early 1960s, it was established that busulfan spontaneously reacts with glutathione and that conjugation with glutathione is the primary route of elimination [27,28]. Studies conducted with various purified human liver GST isoforms established that the highest busulfan-conjugating activity was observed with GSTA1 [19]. Genetic studies described the genomic organization of the human *GST* gene cluster and characterized the functional activity of genetic polymorphisms in the *GSTA1* promoter region [14,15]. From these studies, hypotheses were generated suggesting that decreased functional activity associated with the *GSTA1*B* allele would result in a decreased clearance of busulfan.

In 2006, Kusama et al., investigated for the first time the role of *GSTA1* polymorphisms on busulfan pharmacokinetics in a series of 12 patients [24]. Their results demonstrated that the heterozygous group (*GSTA1*A*B*; *n* = 3) had lower oral clearance, prolonged elimination half-life and higher plasma levels than the wildtype individuals (*GSTA1*A/*A*; *n* = 9). One year later, Kim et al. reported on the first association between *GSTA1* polymorphisms and response to busulfan therapy. [29] To date, very few studies have reexamined the role of *GSTA1* polymorphisms on busulfan pharmacokinetics after oral administration in adult patients. The study by Abbasi et al. reported on a decrease in busulfan clearance in their *GSTA1*B*B* patients’ group treated with oral busulfan while Bremer et al. reported on increased averaged concentration and steady-state (Css) in *GSTA1*B*B* patients [21,25]. The magnitude of changes in busulfan oral clearance observed in our study (20%) in patients with a *GSTA1*B/*B* genotype compared to *GST*A/*A* patients agrees with these results.

The role of *GSTA1* polymorphisms in adults and in children as well as the impact of polymorphisms on other GST isoforms (*GSTM1* or *GSTP1*) on busulfan disposition, effects or toxicity are still controversial [16,21,22,23,25,30,31,32,33,34,35,36,37,38,39]. For instance, Rocha et al. established an association between *GSTP1* and chronic graft vs. host disease but Goekkurt et al. did not observe any correlation between various GST polymorphisms and liver toxicity [30,33]. Following intravenous administration, ten Brink et al., Kim et al. and Choi et al., found a decrease in busulfan clearance ranging from about 12–15% in expresser of the *GSTA1***B* allele while Abbasi et al. found no association [16,23,25,40].

One important observation of our study was that 2/3 of the patients with a *GSTA1*B*B* genotype had mean plasma levels above the upper limit of the therapeutic range (357,140 ng·min/mL) after the first oral dose of busulfan (442,711 ± 46,830 ng·min/mL). A 23% decrease in their subsequent oral doses was required to achieve therapeutic levels. Similar results were observed by Abbasi et al. in their *GSTA1*B*B* patients where a 20% decrease in dose was required between Dose 1 and 5 in order to achieve therapeutic levels [25].

The frequency of the *GSTA1*B*B* observed in our validation healthy subject cohort (20%) was almost identical to the one observed in two other Caucasian populations (20 and 20.8%, respectively) [41,42]. In our adult study cohort receiving busulfan, the **B* variant was found in slightly higher frequency (26% for the **B*B* genotype). An increased frequency of *GSTA1*B*B* expressers was also observed in other Caucasian patients’ population [15,43,44]. The significance of these observations would need to be confirmed in larger studies. 

Finally, we have reported previously on the value of limited sampling strategies (four blood samples) to estimate mean plasma levels of patients undergoing treatment with oral busulfan [5]. This type of approach is of great relevance in sparing blood in patients with leukemia or other blood-related diseases. Bullock et al. also reported very similar results using slightly different time points (4) to calculate AUC [6]. 

## 5. Conclusions

Our study suggests that genetic polymorphisms associated with *GSTA1* explain a significant part of the variability observed for busulfan pharmacokinetics. Our data support the utility of busulfan LSMs strategy clinically and for the interpretation of pharmacogenetics results. 

## Figures and Tables

**Figure 1 pharmaceutics-11-00440-f001:**
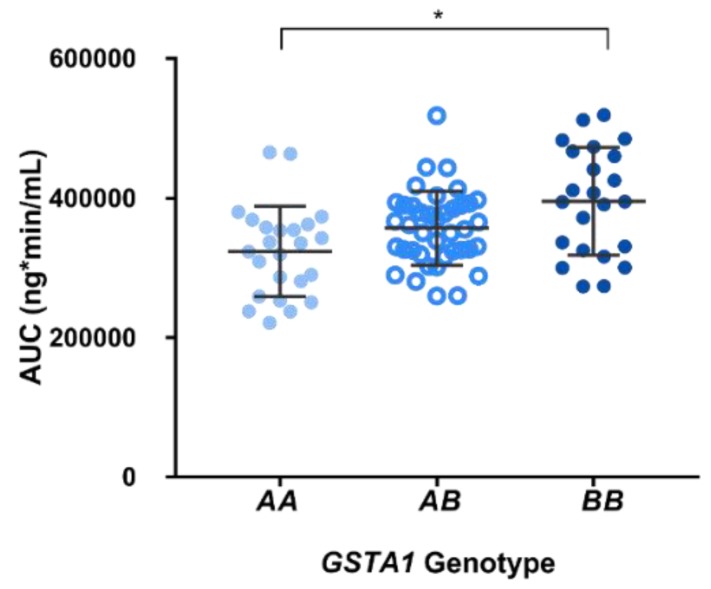
Busulfan plasma concentrations (AUC_0–∞_) measured after administration of the initial oral 1 mg/kg dose (1 mg/kg/day, four times a day, for 4 days) observed among the individual *GSTA1* genotypes for 89 patients enrolled in this study.

**Figure 2 pharmaceutics-11-00440-f002:**
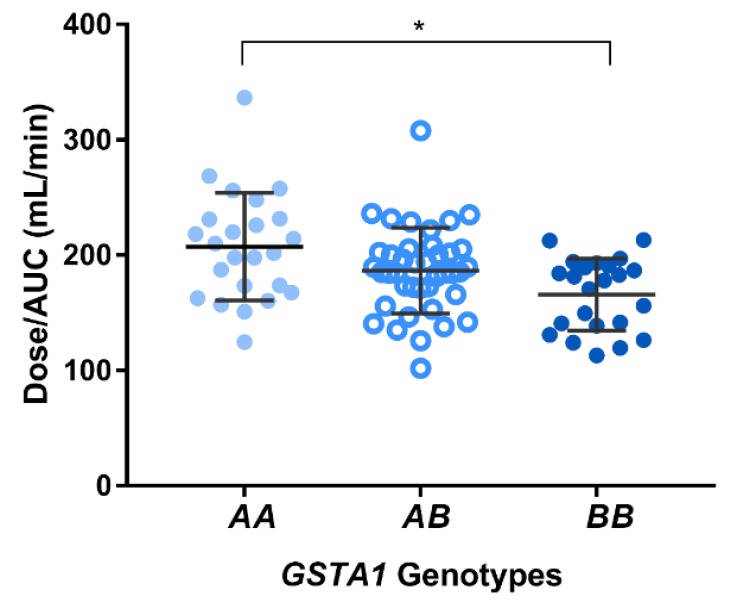
Oral clearance of busulfan calculated after administration of the initial oral dose as a function of patients (*n* = 89) *GSTA1* genotypes.

**Figure 3 pharmaceutics-11-00440-f003:**
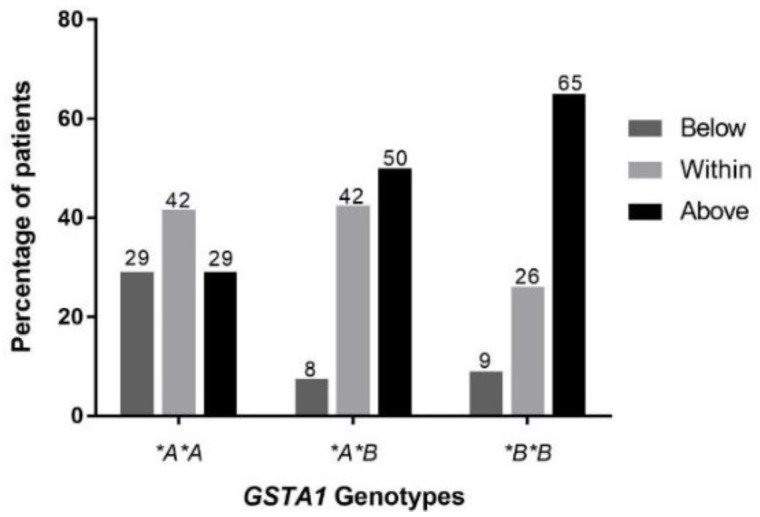
Distribution of patients with an AUC below, within and above the therapeutic range after the initial oral dose of busulfan for each *GSTA1* genotype.

**Figure 4 pharmaceutics-11-00440-f004:**
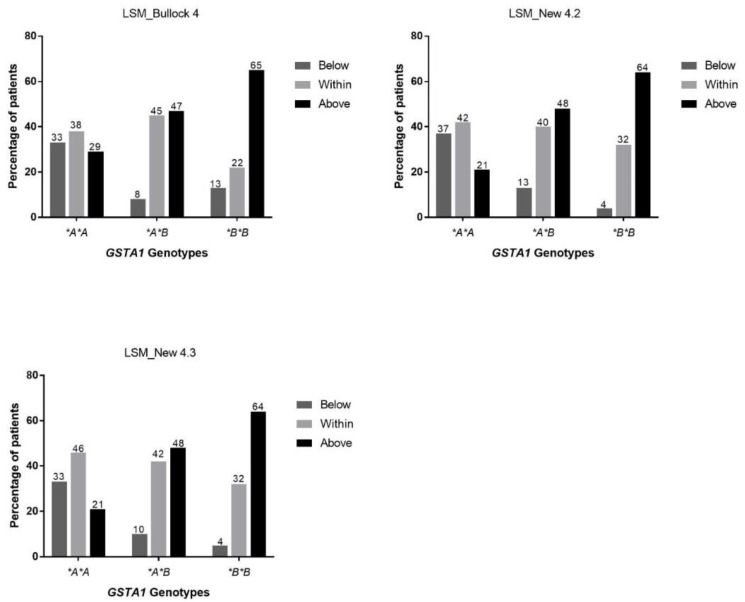
Distribution of patients with an AUC below, within and above the therapeutic range according to their GSTA1 genotype after the first dose of busulfan using 3 limited sampling methods (LSMs) based on 4 blood samples: Bullock 4, New 4.2 and New 4.3.

**Table 1 pharmaceutics-11-00440-t001:** Patient demographics.

Variable	*GSTA1* Genotype Groups			*p*-Value
	**A*A*	**A*B*	**B*B*	
Age: Years ± SD (range)	50 ± 11 (27–65)	48 ± 9 (27–63)	48 ± 10 (25–60)	0.8
Gender: Male/female (% male)	13/11 (54)	26/14 (65)	9/14 (39)	0.4
Weight (Kg)				
Real Body Weight	74 ± 11	73 ± 15	76 ± 19	0.8
Adjusted Ideal Body Weight	65 ± 9	66 ± 11	64 ± 11	0.7
Lean Body Weight	64 ± 9	64 ± 10	61 ± 10	0.3
Bilirubin (U/L)	11 ± 6	14 ± 10	10 ± 5	0.2
AST (U/L)	22 ± 10	24 ± 9	22 ± 11	0.7
ALT (U/L)	27 ± 22	34 ± 23	33 ± 34	0.5
Albumin (g/L)	41 ± 4	42 ± 3	43 ± 5	0.3
Alkaline Phosphatase (U/L)	95 ± 38	86 ± 36*	81 ± 23	0.01
LDH (U/L)	280 ± 285*	166 ± 59	169 ± 43	0.01
Previously received chemotherapy (%)	22 (92)	35 (88)	19 (83)	0.2
Previously received radiotherapy (%)	3 (13)	4 (10)	2 (13)	0.8
Number of patients taking Acetaminophen (%)	9 (37)	8 (20)	6 (26)	0.02
Number of patients taking Antifungal Drugs (%)	2 (8)	3 (7)	1 (4)	0.4
First dose administered (mg)	65 ± 8	66 ± 12	65 ± 14	0.9

* Tukey’s multiple comparison analysis, the group (*) was statistically different vs. the 2 other genotype groups.

**Table 2 pharmaceutics-11-00440-t002:** *GSTA1* genotype frequencies.

Patients/Cohort	*n*	*GSTA1* Genotypes % (*n*)		
		**A*A*	**A*B*	**B*B*
Adult patients treated at HRM (study population)	87	27.6% (24)	46% (40)	26.4% (23)
Healthy man subjects (validation cohort)	116	31% (36)	49% (57)	20% (23)

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
