# Peer review of "Impact of GSTA1 Polymorphisms on Busulfan Oral Clearance in Adult Patients Undergoing Hematopoietic Stem Cell Transplantation"

_pharmaceutics, 2019, doi:10.3390/pharmaceutics11090440_

Round 1

Reviewer 1 Report

In the current study, the authors evaluate the influence of GSTA1 gene variants on busulfan oral clearance in a population of patients undergoing hematopoietic stem cell transplantation. They found patients carrying GSTA1 loss of function *B allele were at increased risk of overdosing on their initial busulfan oral dose.

Overall, the study is plausible, thus, several flaws exist.

1. Why only GSTA1 gene was chose, why not also including GSTT1, GSTM1 and GSTP1 genes?

2. What about the response rate for the including subjects?

3. Results for PCR-RFLP should be validated at least 10% samples by direct sequencing.

4. Why genetic analyses were performed only in 89 patients not all of the 119 patients?

Author Response

Responses to reviewers

Reviewer #1

Comments and Suggestions for Authors

In the current study, the authors evaluate the influence of GSTA1 gene variants on busulfan oral clearance in a population of patients undergoing hematopoietic stem cell transplantation. They found patients carrying GSTA1 loss of function *B allele were at increased risk of overdosing on their initial busulfan oral dose.

Overall, the study is plausible, thus, several flaws exist.

Why only GSTA1 gene was chose, why not also including GSTT1, GSTM1 and GSTP1 genes?

We have concentrated and restricted our analyses to GSTA1 gene polymorphisms for several reasons:

Busulfan is extensively metabolized following administration in human with less than 2% of the dose is excreted as unchanged busulfan in the urine. The primary elimination pathway for busulfan involves glutathione (GSH) conjugation. Two distinct superfamilies of GST isoenzymes exist: cytosolic/soluble (dimeric) which are involved in drug metabolism and microsomal (trimeric) involved in arachidonic acid metabolism. Gibbs et al. showed that human liver cytosolic, but not microsomal, GST enzymes metabolize busulfan to the sulfonium ion of GSH. This pathway accounted for 50% of its metabolism.1 The human cytosolic/soluble GST superfamily is subdivided into eight separate classes designated Alpha, Kappa, Mu, Pi, Sigma, Theta, Zeta and Omega. By contrast with other members of this superfamily, the class Kappa is mitochondrial. The isoform GSTA1 is the predominant GSH catalyzing busulfan conjugation in human liver. Whereas GSTM1 and GSTP1 could also catalyze however, their contribution is considered minor as in vitro experiments showed that GSTM1 and GSTP1 had 46% and 18% of the intrinsic clearance of GSTA1. In the intestine, conjugation of busulfan is also catalyzed by GSTA1 with an activity similar to the hepatic GSTA1.1-3 Expression of various isoforms of GST is tissue-specific. GSTA1 is much more abundant in the liver and small intestine than other isoenzymes. For instance, in the liver, it has been shown to be at least 3.5-fold more abundant than GSTM1 and at least 60-fold more abundant than GSTP1. The relative contribution of these isoenzymes to the overall conjugation of busulfan to GSH metabolism of busulfan is expected to be minimal.4 For these reasons, our study has focused on the role of GSTA1 polymorphism on the disposition of busulfan.

What about the response rate for the including subjects?

Data collected in the course of this study are limited to pharmacokinetics (plasma concentrations) and demographic data. We did not have response rate, adverse events or any follow-up over time. The association with clinical outcome should further be investigated in a long-term study.

Results for PCR-RFLP should be validated at least 10% samples by direct sequencing.

The genotype method used in our study has been developed and validated by Kusama et al. The GSTA1 genotype was not used as a commercial pharmacogenetics-based decision dosing tool. In our laboratory, the method has been validated using a cohort of 116 random samples, using good laboratory practices, and making sure that the method could reproduce Hardy-Weinberg equilibrium conditions.

Why genetic analyses were performed only in 89 patients not all of the 119 patients?

Not all patients provided their consent for genetic analysis and a full pharmacokinetics profile was also missing for few patients, thus we have complete dataset for 89 patients.

References

Gibbs JP, Czerwinski M, Slattery JT. Busulfan-glutathione conjugation catalyzed by human liver cytosolic glutathione S-transferases. Cancer Res. 1996 Aug 15; 56(16):3678-81. Hassan M, Oberg G, Ehrsson H, Ehrnebo M, Wallin I, Smedmyr B, Tötterman T, Eksborg S, Simonsson B. Pharmacokinetic and metabolic studies of high-dose busulphan in adults. Eur J Clin Pharmacol. 1989; 36(5):525-30. Czerwinski M, Gibbs JP, Slattery JT. Busulfan conjugation by glutathione S-transferases alpha, mu, and pi. Drug metabolism and disposition: the biological fate of chemicals (1996). Rowe JD, Nieves E, Listowski I. Subunit diversity and tissue distribution of human glutathione S-transferases: interpretations based on electrospray ionization-MS and peptide sequence-specific antisera. Biochem j 1997;325:481-486.

Reviewer 2 Report

This is a nice paper showing the impact of GSTA1 polymorphisms on busulfan oral clearance in adult patients undergoing hematopoietic stem cell transplantation. I only have some minor criticisms:

Has this study been registered? Figures should be understandable without references to the main text. Please correct the legends of figures so that they are legible. Figure S1 has no legend at all.

Author Response

Reviewer #2

Comments and Suggestions for Authors

This is a nice paper showing the impact of GSTA1 polymorphisms on busulfan oral clearance in adult patients undergoing hematopoietic stem cell transplantation. I only have some minor criticisms:

Has this study been registered?

The study was not registered at the NIH website. The study was reviewed by the local IRB (No 06068).

Figures should be understandable without references to the main text. Please correct the legends of figures so that they are legible. Figure S1 has no legend at all.

We thank this reviewer for these comments. Please note that unfortunately, information presented in this version of the manuscript does not correspond to information provided to the editor.

We have added appropriate legends as requested.

Figure 1. Busulfan plasma concentrations (AUC0-∞) measured after administration of the initial oral 1mg/kg dose (1mg/kg/day, four times a day, for 4 days) observed among the individual GSTA1 genotypes for 89 patients enrolled in this study.

Figure 2. Oral clearance of busulfan calculated after administration of the initial oral dose as a function of patients (n=89) GSTA1 genotypes.

Figure 3. Distribution of patients with an AUC below, within and above the therapeutic range after the initial oral dose of busulfan for each GSTA1 genotype.

Figure 4. Distribution of patients with an AUC below, within and above the therapeutic range according to their GSTA1 genotype after the first dose of busulfan using 3 limited sampling methods (LSMs) based on 4 blood samples: Bullock 4, New 4.2 and New 4.3.

Supplemental Figure S1. Percentage of patients with an AUC below, within and above the therapeutic range after the initial oral dose of busulfan using reference AUC vs. 3 limited sampling methods (LSMs) based on 4 blood samples: Bullock 4, New 4.2 and New 4.3.

Round 2

Reviewer 1 Report

The response are acceptable.